# Biaxial strength under fatigue of 4YSZ: Surface treatment and resin cement viscosity effects

Kétlin Fagundes Teixeira[1], Fernanda Cocco[1], Pablo Machado Soares[1], Gabriel Kalil Rocha Pereira[1], Cornelis Johannes Kleverlaan[2], João Paulo Mendes Tribst[3]*, Luiz Felipe Valandro[1], Ana Carolina Cadore-Rodrigues[1]

1 MSciD-PhD Post-Graduate Program in Oral Science, Faculty of Odontology, Federal University of Santa Maria, Santa Maria, Rio Grande do Sul, Brazil, 2 Department of Dental Materials Science, Academic Centre for Dentistry Amsterdam (ACTA), Universiteit van Amsterdam and Vrije Universiteit, Amsterdam, Noord-Holland, The Netherlands, 3 Department of Reconstructive Oral Care, Academic Centre for Dentistry Amsterdam (ACTA), Universiteit van Amsterdam and Vrije Universiteit, Amsterdam, Noord-Holland, The Netherlands

* j.p.mendes.tribst@acta.nl

## Abstract

This study aimed to investigate the effect of different surface treatments combined with high- or low-viscosity resin cements on the biaxial flexural strength of 4YSZ ceramics. Discs of 4YSZ (IPS e.max ZirCAD MT, Ivoclar AG; 15 mm diameter, 1.2 mm thickness) were prepared, subjected to CAD/CAM milling simulation using #220 silicon carbide papers, and divided into 10 groups according to two factors: "Surface Treatment" – Control (CTRL, no treatment); air abrasion with 45 μm (AB45) or 110 μm (AB110) alumina oxide particles; and glaze either followed (GLZ+HF) or not (GLZ) by 10% hydrofluoric acid; and "Resin Cement Viscosity" – High (H) or Low (L). The discs were treated, luted with resin cement, and subjected to two mono-tonic tests (n = 2) to determine fatigue parameters, followed by biaxial fatigue testing (n = 15; piston-on-three-balls; 20 Hz; 10,000 cycles; initial stress: 75 MPa; increments of 25 MPa). Topography, fractography, and surface roughness analyses were also performed. Surface treatment significantly influenced fatigue performance (p < 0.05, F = 60.25), whereas resin cement viscosity and its interaction with surface treatment did not show significant effects (p > 0.05, F = 0.86; p > 0.05, F = 0.07). The GLZ groups exhibited the highest flexural strength (H/L-GLZ: 382–396 MPa), followed by interme-diate values in CTRL (H/L-CTRL: 285–288 MPa), AB45 (H/L-AB45: 310–315 MPa), and GLZ+HF (H/L: 273–280 MPa), while AB110 groups presented the lowest strength (H/L: 205–210 MPa). Surface roughness was lower for GLZ (Ra: 1.44 μm) and GLZ+HF (Ra: 1.38 μm) compared to AB45 (Ra: 1.83 μm) and AB110 (Ra: 1.69 μm). GLZ produced smooth and homogeneous surfaces, GLZ+HF led to areas partially exposed by acid, and AB110 caused more pronounced surface defects than AB45. Hence, the GLZ treatment offered superior fatigue performance; AB45 and

**Data availability statement:** All relevant data are within the manuscript and its Supporting Information files.

**Funding:** This work is part of the fulfillment of the requirements for the MSci degree (K.F.T.) in the Post-Graduate Program in Oral Sciences at the Faculty of Dentistry, Federal University of Santa Maria (Rio Grande do Sul State, Brazil). This research was partially financed by the Brazilian Federal Agency for Coordination of Improvement of Higher Education Personnel – CAPES (Finance Code 001 for K.F.T. Master's scholarship); by the Research Support Foundation of the State of Rio Grande do Sul – FAPERGS (#2551-0000152-7/2023 for A.C.R. Post-doctoral scholarship); and by the Brazilian National Council for Scientific and Technological Development – CNPq (#304665/2022-3 for G.K.R.P research grant; and #308427/2021-1 for L.F.V research grant; #442272/2023-5). Furthermore, we extend our heartfelt gratitude to Ivoclar AG in providing some of the materials used in this research. It is important to emphasize that the contribution of these institutions was limited to financial support, with no involvement in the study's design, data collection or analysis, publication decisions, or manuscript preparation.

**Competing interests:** The authors have declared that no competing interests exist.

GLZ + HF showed behavior similar to CTRL, whereas AB110 impaired mechanical performance. Resin cement viscosity had no significant influence on fatigue strength

---

## 1. Introduction

Zirconia-based ceramics have gained significant attention and are widely used in clinical practice for posterior dental crowns due to their excellent mechanical properties, particularly high toughness and biocompatibility [1–4]. To extend their use to anterior restorations, an increased cubic phase content was introduced, enhancing optical translucency while maintaining satisfactory mechanical performance [5,6]. Thereby, zirconia with 4% yttria content (4YSZ) was developed for inlays, onlays, veneers, and crowns [7]. However, the higher cubic content limits tetragonal-to-monoclinic phase transformation, reducing the material's ability to resist crack propagation and potentially compromising the mechanical performance of 4YSZ restorations [5,6,8,9].

Furthermore, considering that zirconia ceramics are polycrystalline and silica-free, their surfaces are intrinsically unreactive to hydrofluoric acid [10,11], which limits direct chemical bonding with resin cements and necessitates alternative surface treatments. In this context, air abrasion with alumina oxide, with or without silica, remains the most commonly used approach [12–14]. Alumina air abrasion erodes the ceramic surface, creating topographical irregularities and increasing surface roughness [15–20], thereby enhancing mechanical interlocking with resin cement [17,21]. When silica particles are incorporated, air abrasion additionally deposits a silica layer on the zirconia surface, facilitating chemical interaction with silane and resin cement [17,20,22,23].

Regarding air abrasion protocols, a systematic review by Aurélio et al. [24] indicated that particle size and composition applied on 3 mol% yttria-stabilized zirconia (3Y-TZP) do not produce distinct mechanical behaviors. Both 45 µm and 110 µm particles induced similar flexural strength, suggesting that larger particles alone do not compromise mechanical performance. The main reason for the lack of difference is that, due to the phase transformation in 3Y-TZP, both particle sizes induce transformation, resulting in similar strength. However, the effect of particle size and type on higher-translucency zirconia, such as 4YSZ, is less understood. Smaller particles (45 µm) tend to produce less severe surface defects, better preserving mechanical properties, whereas larger particles (110 µm) may generate deeper flaws acting as stress concentrators [24,25].

To address these challenges, various alternative surface treatments have been proposed. Among them, the application of a thin layer of low-fusion glass ceramic rich in silica (glaze) has been shown to smooth surface irregularities and partially fill defects generated during material processing, thereby enhancing surface uniformity and overall material stability [25–27]. In addition, glaze deposition improves adhesion to silane and resin cement, as demonstrated in bond strength studies [28,29]. The subsequent use of hydrofluoric acid (HF) after glazing remains controversial: some studies apply silane alone [25,30], whereas others combine HF etching with silanization [29,31]. Notably, most evidence pertains to 3Y-TZP, and data regarding the

effects of HF on 4YSZ are scarce, emphasizing the need to investigate whether surface modification after glaze application should involve acid etching or silane treatment alone.

Given that the overall mechanical performance of zirconia restorations depends not only on surface treatment but also on various factors, resin cement properties are equally critical. Effective interpenetration onto the surface irregularities increases the energy required for failure, enhancing structural performance. Cement viscosity might affect it: low-viscosity cements penetrate shallow defects more effectively, while high-viscosity cements, often with higher filler content, support higher load-bearing capacity. Aragonez et al. [32] evaluated fatigue behavior of 4YSZ using high- and low-viscosity cements combined with alumina oxide air abrasion, finding no significant differences. Nevertheless, since air abrasion may alter or damage 4YSZ surfaces, affecting mechanical properties due to limited phase transformation and the absence of the "protective" tetragonal phase [5,7,9].

Thus, it is essential to investigate how different surface treatments, which produce distinct surface characteristics, interact with resin cement viscosity to determine their combined effect on mechanical performance. The novelty of this study lies in the combined evaluation of multiple surface treatments and two resin cement viscosities on the fatigue biaxial flexural strength of 4YSZ. Unlike previous studies that have predominantly investigated surface treatments or resin cement viscosity separately, this work integrates both factors within the same experimental framework, allowing a direct assessment of their combined effect on mechanical performance. To the best of the authors' knowledge, no previous study has evaluated the interaction between surface treatment and resin cement viscosity on the fatigue biaxial flexural strength of 4YSZ. Therefore, this study aimed to examine the effect of different surface treatments followed by adhering to the high- or low-viscosity resin cements on the biaxial flexural strength under fatigue of a 4YSZ ceramic. The hypotheses formulated were: (1) Surface treatment affects the biaxial flexural strength of 4YSZ; (2) Resin cement viscosity affects the biaxial flexural strength of 4YSZ.

## 2. Materials and methods

The Table 1 provides a comprehensive overview of the materials used in this study, including their manufacturers, batch numbers, and compositions.

### 2.1. Study design

For the fatigue testing, 10 groups (n = 15) were designed considering two factors:

- *Surface Treatment*: Control – no treatment (CTRL); air abrasion with 45 μm (AB45) and 110 μm (AB110) alumina oxide particles; glaze spray followed (GLZ + HF) or not (GLZ) by 10% hydrofluoric acid;

- *Resin Cement Viscosity*: high (H) and low (L).

The number of specimens per group was determined based on previous studies published in the literature that evaluated dental ceramics under the concepts of fatigue and flexural strength [33–36] and was also supported by a sample size power calculation, which confirmed adequate statistical sensitivity with a power of 100%. The study flowchart is presented in Fig 1.

### 2.2. Specimen preparation

**2.2.1. Ceramic discs.** The specimens were prepared following ISO 6872:2024 standards for biaxial flexural strength testing. Initially, 4YSZ blocks (IPS e.max ZirCAD MT, Ivoclar AG, Schaan, Liechtenstein) were manually sectioned into smaller pieces measuring 20 mm × 20 mm. These pieces were positioned between two metallic guides 18 mm and ground using a polishing machine (Ecomet/Auto-met 250, Buehler; Lake Bluff, USA) with #400, #600 and #1200-grit silicon carbide papers (Norton, Saint-Gobain Abrasives; São Paulo, Brazil) under continuous water cooling until they achieved

**Table 1. List of materials used in the study: commercial name, manufacturer, batch number and composition based on the manufacturer's information.**

| Commercial name | Manufacturer (batch number) | Composition |
|---|---|---|
| IPS e.max ZirCAD MT | Ivoclar AG (Y34224) | $86.0 \leq 93.5\%$ $ZrO_2$; $6.5 \leq 8.0\%$ $Y_2O_3$; $\leq 5.0\%$ $HfO_2$; $\leq 1.0\%$ $Al_2O_3$ |
| Variolink N Low viscosity catalyst | Ivoclar AG (Z00XG) | Barium glass filler, consisting of mixed oxide (52.2 wt%); Dimethacrylates (22.0 wt%); Ytterbium Trifluoride (25.0 wt%); along with a small portion of Initiators and stabilizers (0.8 wt%) and Pigments (<0.1 wt%). |
| Variolink N High viscosity catalyst | Ivoclar AG (Z010VB (Z010W) | Barium glass filler, mixed oxide (46,2 wt%); Dimethacrylates (27,9 wt%); Ytterbium Trifluoride (25,0 wt%); Initiators, stabilizers (0,9 wt%); Pigments (<0,1 wt%) |
| Variolink N Base | Ivoclar AG Z00TZM (Z0040FV) | Barium glass filler, mixed oxide (48,4 wt%); Dimethacrylates (26,3 wt%); Ytterbium Trifluoride (25,0 wt%); Initiators, stabilizers (0,3 wt%); Pigments (<0,1 wt%) |
| 10% hydrofluoric acid | Condac (310724) | 10% concentration hydrofluoric acid |
| VITA Akzent Plus | VITA Zahnfabrik (E78440) | Amorphous glassy substance (silica-based material) |
| Monobond N | Ivoclar AG (Y45831) | Alcohol solution of silane methacrylate, phosphoric acid methacrylate and sulphide methacrylate. |

a cylindrical shape with a diameter of 18 mm. The cylinders were then cut in discs using a diamond cutting disc attached to a precision cutting machine (IsoMet 1000, Buehler) under water cooling. Finally, the discs were polished on both sides using #600 and #1200-grit silicon carbide papers under water cooling to eliminate any irregularities from the cutting process, resulting in a final pre-sintering thickness of 1.6 mm.

**2.2.2. CAD/CAM milling simulation.** For the CAD/CAM milling simulation, the pre-sintering roughness was expected to be 2.2 µm and the post-sintering roughness 1.8 µm [37,38]. For this, the discs were ground on #220-grit silicon carbide papers (5 x 10 cm for each disc) applying light digital pressure for 2 seconds in two movements (up and down) along the x and y axes to achieve the desired surface roughness [37,38]. To ensure standardization, a contact profilometer (Mitutoyo SJ 400 Profilometer, Mitutoyo Corporation, Kawasaki, Japan) was used to measure the roughness of the ceramic specimens both before and after sintering. The pre-sintering roughness off all specimens had a mean Ra value of 2.27 µm. The specimens were then sintered according to the manufacturer's recommendations, with final dimensions of 14 mm in diameter and 1.2 mm in thickness. After sintering, the roughness of all specimens was measured again, presenting a final mean Ra value of 1.82 µm. Then, the specimens were randomized and distributed according to the study design (Fig 1).

## 2.3. Surface treatments

The specimens underwent cleaning in an ultrasonic bath (1440 D – Odontobras, Ind. E Com. Equip. Méd. Odonto. LTDA, São Paulo, Brazil) with distilled water for 5 minutes. The intaglio surfaces of the 4YSZ discs were subjected to the respective surface treatments under study, as described in Fig 1. The surface treatments were performed by a trained operator to ensure standardization. The thickness of the glaze was measured before and after the application using a digital caliper (Digital Caliper, Mitutoyo Absolute 500–196−20 Digital) to assess the variation in glaze layer thickness, given that its

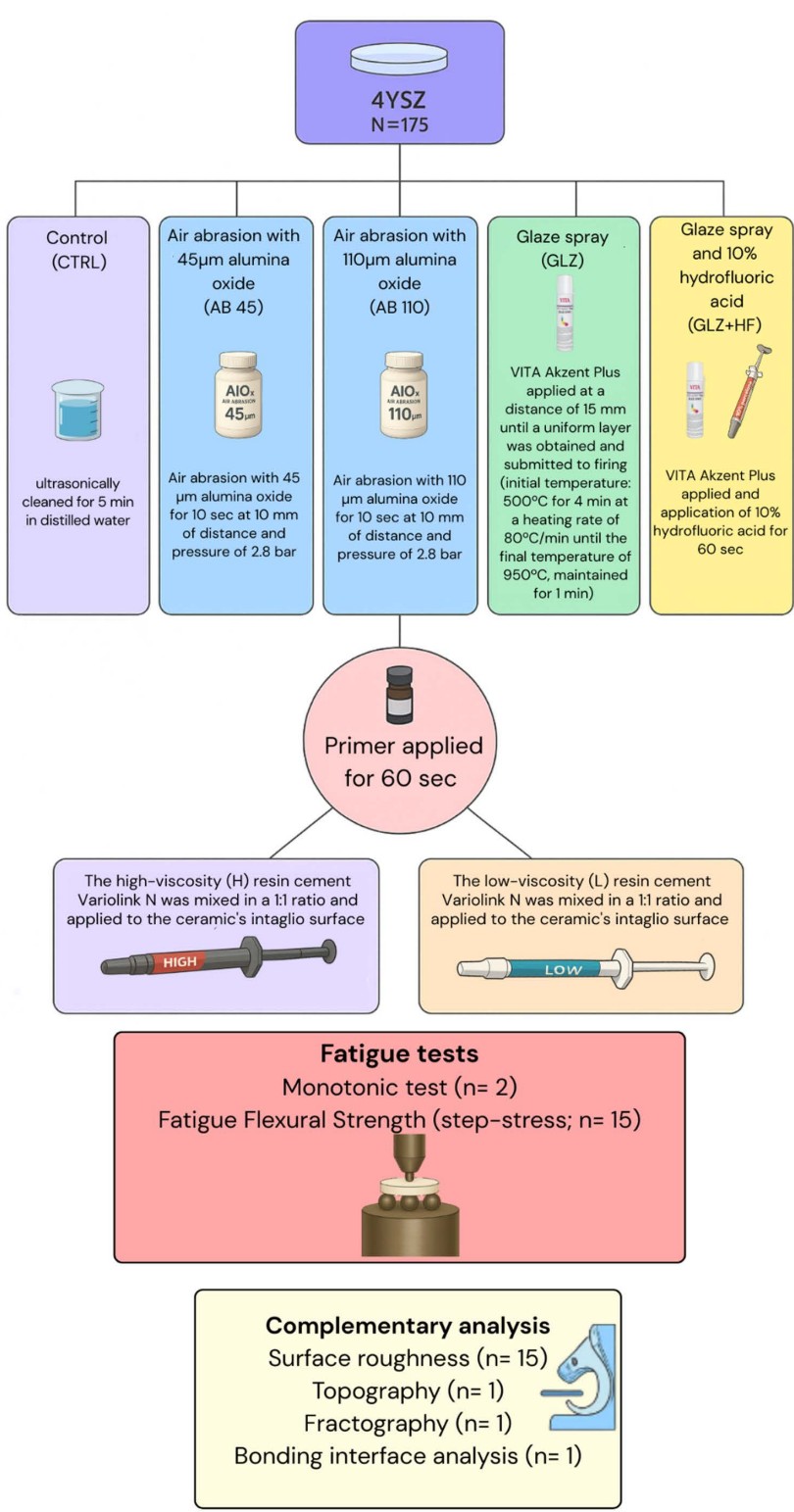

**Fig 1. Flow diagram of study design.**

application may result in an increase of approximately 16 µm, the overall average thickness across all treated groups is presented based on the individual group means shown in Fig 2.

## 2.4. Surface roughness analysis

A micrometric analysis was conducted on all specimens (n = 15) using a contact profilometer (Mitutoyo SJ 400 Profilometer, Mitutoyo Corporation) after the surface treatments had been applied. This analysis aimed to assess the surface roughness of the groups, with four measurements performed on each specimen, considering the Ra and Rz parameters in accordance with ISO 21920:2021. Specifically, two measurements were taken along the x-axis and two along the y-axis. Ra, representing the arithmetical mean of the absolute values of peaks and valleys measured from a mean plane (in µm), and Rz, representing the average distance between the five highest peaks and five deepest valleys of a surface (in µm), were evaluated

## 2.5. Luting procedure

Subsequently, a primer agent (Monobond N, Ivoclar AG) was applied on the intaglio surface of the ceramic discs treated for 60 seconds.

The dual resin cement (Variolink N, Ivoclar AG) was mixed in a 1:1 ratio. A standard volume of each cement high (H) or low (L) viscosity was applied to the center of the treated ceramic surface. Thereafter, the disc, coated with a thin layer of resin cement, was placed on a thin polyester strip attached to a glass slide [39]. After positioning the assembly centrally on a loading platform, the upper ceramic surface was loaded with 2.5 N for 5 minutes to standardize the cement thickness [39]. Excess resin cement was removed with a microbrush. Light activation (1200 mW/cm2; Radii-cal LED curing light, SDI; Bayswater, Australia) was performed for 5 exposures of 20 seconds each (0°, 90°, 180°, 270°, and top). All cemented specimens were stored in distilled water at an oven (37°C) (at least 24 hours until 7 days) until monotonic (n = 2) and fatigue (n = 15) tests were conducted.

After the application of the resin cement, the thickness of the cement layer was determined using a digital caliper (Mitutoyo Absolute 500–196−20 Digital Caliper). The cement layer thickness was determined by measuring the difference in specimen thickness before and after cementation. To ensure consistency and minimize its impact on mechanical performance, specimens with values outside the ~50 µm range were excluded, in accordance with ISO 4049:2000, the

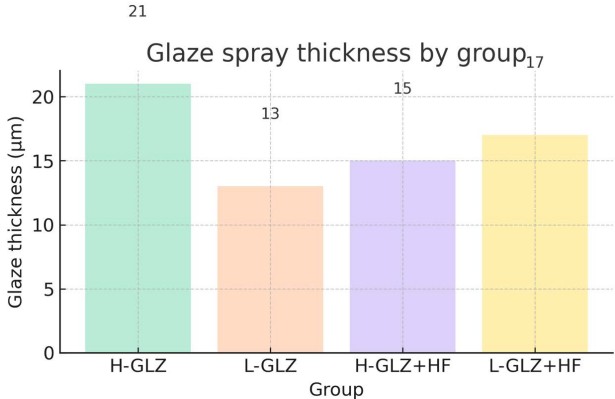

**Fig 2. Average glaze layer thickness for each treated group.** The figure shows the individual group means, representing the increase in surface thickness due to glaze application.

average thickness values for each group are shown in Fig 3. This approach ensured a standardized monolayer structure, as previously described by Packaeser et al. [39]. Thus, a ceramic–cement configuration was obtained.

## 2.6. Monotonic test

A preliminary monotonic test was conducted using a DL-2000 Emic machine (Instron, São José dos Pinhais, Brazil). The specimens (n = 2) were positioned with the treated surface facing downward (tensile side) on three support balls (Ø = 3 mm) under distilled water. A flat circular tungsten piston (Ø = 1.6 mm) was applied perpendicularly to the center of each disc (ISO 6872: 2024). Prior to testing, adhesive tape was applied to the compression side of the discs to prevent fragment dispersion and ensure proper contact between the piston and the sample. A cellophane tape (110 μm) was positioned between the specimens and the support balls to improve the stress distribution and reduce the risk of contact damages on the tensile side [40].

The resulting data (in newtons) were used to calculate the flexural strength (in MPa), using the following formula:

$$\sigma_{BI} = \frac{-0,2837P(X-Y)}{b^2}$$

Equation 1

In which: $\sigma_{BI}$ was the maximum center tensile stress, in Megapascals; $P$ was the total load causing fracture (in Newtons); $b$ was the specimen's thickness (in mm). Also, the coefficients $X$ and $Y$ were determined:

$$X = (1 + \nu)\ln\left(\frac{r2}{r3}\right) + \left(\frac{1-\nu}{2}\right)\left(\frac{r2}{r3}\right)$$

Equation 2

$$Y = (1 + \nu)\left[1 + \ln\left(\frac{r1}{r3}\right)\right] + (1 - \nu)\left(\frac{r1}{r3}\right)$$

Equation 3

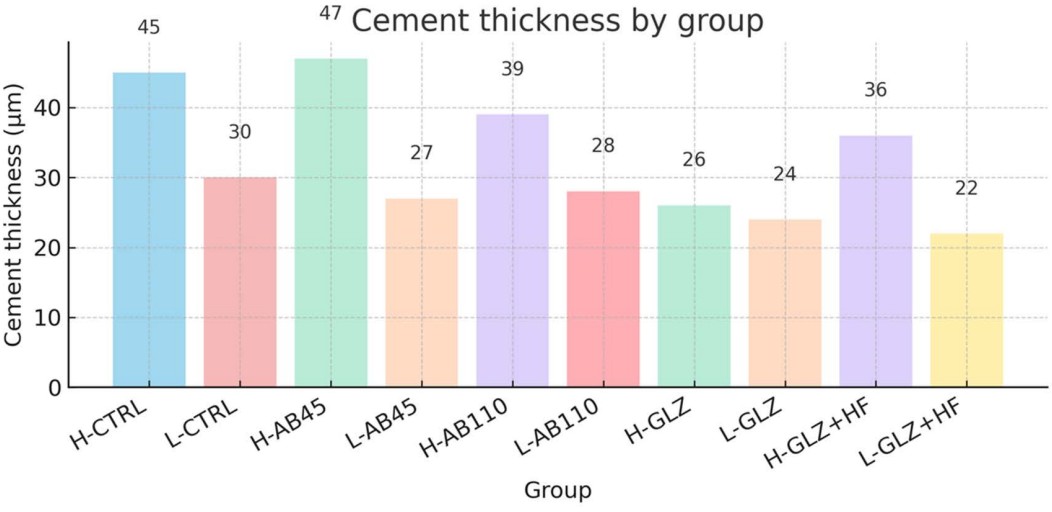

**Fig 3. Illustration of the ceramic–cement configuration used in this study, with cement layer thickness determined by the difference in specimen thickness before and after cementation.**

Where: $\nu$ was the Poisson's ratio, $r1$ was the radius of the support circle (in mm), $r2$ was the radius of the piston (in mm), and $r3$ was the radius of the specimen (in mm).

The average fracture load strength (463.65 MPa) obtained in the monotonic test was used as parameters for the fatigue test.

## 2.7. Fatigue test

A mechanical testing machine (Instron Corporation; Norwood, USA) was used to conduct the cyclic fatigue test (n = 15) using the piston on-three-balls setup and methods previously described (item 2.5).

Initially, a cyclic load was applied using a tungsten cylindrical piston with a diameter of 1.6 mm at a frequency of 20 Hz, following the protocol described by Fraga et al. [41]. The initial stress was applied at 75 MPa (~15% of the monotonic test) and subsequently, increments of 25 MPa (~5% of the monotonic test) were applied at each 10,000 cycles until failure occurred. The test was conducted with the specimens immersed in distilled water. After fracture, the fatigue strength (FS) and cycles to failure (CFF) data were recorded for statistical analysis.

## 2.8 Scanning electron microscopy (SEM)

For the scanning electron microscopy (SEM) evaluation, ceramic specimens prepared for each surface treatment (n = 1) were ultrasonically cleaned (1440 D – Odontobras) with distilled water for 5 minutes and examined at 500 × magnification (Carl Zeiss, Oberkochen, Baden-Württemberg, Germany) to assess surface topography. The air-abrasion powders (45 μm and 100 μm) were subsequently mounted, sputter-coated with a thin gold–palladium layer, and observed at 200 ×, 500 ×, and 1000 × magnifications to characterize their morphology (VEGA3, Tescan, Brno, Czech Republic).

To evaluate resin cement infiltration, one specimen from each selected group (H-CTRL and L-CTRL) was sectioned to expose the intaglio surface, and the ceramic/cement/substrate assemblies were mirror-polished with #1200 and #2400 SiC papers (EcoMet/AutoMet 250, Buehler), ultrasonically cleaned in distilled water for 5 minutes, metallized, and examined by SEM (VEGA3, Tescan).

Finally, all fractured specimens were initially inspected under an optical stereomicroscope (Stereo Discovery V20, Carl Zeiss, Gottingen, Germany) to identify the origin of failure. Representative samples (n = 1) were then ultrasonically cleaned, air-dried, metallized, and analyzed under SEM (VEGA3, Tescan) at 200× and 1000 × magnifications.

## 2.9 Data analysis

Since the roughness data exhibited a parametric and homoscedastic distribution, as indicated by Shapiro-Wilk and Levene tests (p > 0.05), One-way ANOVA and Tukey's post-hoc tests ($\alpha$ = 0.05) were conducted using SPSS version 21 (IBM Analytics, New York, United States) for the analysis following surface treatments.

Statistical analysis of survival rates, flexural strength (FS), and cycles to failure (CFF) was performed using Kaplan-Meier and Mantel-Cox (Log-Rank) tests ($\alpha$ = 0.05; SPSS version 21, IBM Analytics). FS and CFF data were also analyzed by Two-Way ANOVA to evaluate the effects of resin cement viscosity (high vs. low) and surface treatment, including possible interactions between factors. Additionally, Weibull analysis was conducted using SuperSMITH Weibull 4.0k-32 software (Wes Fulton, Torrance, United States) to characterize the distribution and reliability of fatigue data. Significant differences in the Weibull modulus (m) were determined using the maximum-likelihood method, providing insight into the mechanical reliability of each condition.

A sample size and power analysis was conducted using G*Power software (version 3.1.9.6, Germany) to estimate the statistical power of the study. The calculation employed an F-test for ANOVA with fixed effects, including main effects and interactions, at an alpha level of 0.05. A total sample size of 150 and an effect size f of 1.503887, derived from the FS means and their corresponding standard deviations, were used. Statistical analyses for the different outcomes were

performed using IBM SPSS (version 21, IBM) and Statistix 10 (Analytical Software, USA), with a significance threshold set at α = 0.05. The achieved power was calculated with a noncentrality parameter λ = 339.2514, critical F = 1.9473, numerator degrees of freedom = 9, denominator degrees of freedom = 140, resulting in a power of 1–β = 1.00 (100%).

The SEM analyses were conducted in a subjective and descriptive manner, with the aim of observing and interpreting the effects of the different surface treatments on the specimens and their interaction with the viscosity of the resin cement.

## 3. Results

According to the results obtained through Two-way ANOVA, surface treatments showed a significant effect on flexural strength (FS) or cycles to failure (CFF) ($p < 0.05$, F = 60.25). Table 2 shows that the H-GLZ and L-GLZ groups exhibited the highest FS and CFF, with statistically similar results. On the other hand, the L-CTRL, H-CTRL, L-AB45, H-AB45, GLZ + HF, and H-GLZ + HF groups showed intermediate performance, while L-AB110 and H-AB110 had lower FS and CFF values. Therefore, surface treatment is the determining factor in mechanical behavior (H-GLZ = L-GLZ > H-CTRL = L-CTRL = H-AB45 = L-AB45 = H-GLZ + HF = L-GLZ + HF > H-AB110 = L-AB110). The Weibull modulus showed statistically similar results, suggesting equivalent structural reliability among all the groups analyzed (Table 2). However, the viscosity of the resin cement and the interaction between the resin cement and surface treatments did not have a significant effect on fatigue behavior (FS and CFF) ($p > 0.05$, F = 0.86; $p > 0.05$, F = 0.07, respectively). These data indicate that the viscosity of the resin cement did not significantly alter the effect of surface treatments on the fatigue strength of the materials (H-CTRL vs. L-CTRL p = 0.70; H-AB45 vs. L-AB45 p = 0.21; H-AB110 vs. L-AB110 p = 0.12; H-GLZ vs. L-GLZ p = 0.54; H-GLZ + HF vs. L-GLZ + HF p = 0.83).

Moreover, bonding interface analysis presented in Fig 4 supported these findings, showing no significant differences in cement layer integrity or adaptation between specimens cemented with high viscosity and low viscosity resin cements. This suggests that resin cement viscosity does not influence the quality of the cementation interface, which correlates with the absence of significant differences in mechanical fatigue behavior between these groups.

In Table 3, differences in survival rates for 250 MPa the H-GLZ group showed a high survival rate of 93%, while the H-GLZ + HF group had a survival rate of 40%, indicating that the application of HF after glazing reduces the fatigue resistance of 4YSZ. The H-AB45 group also showed a survival rate of 93%, whereas the H-AB110 group exhibited a lower rate

**Table 2. Results of fatigue data by means of Kaplan-Meier and Mantel-Cox (Log-Rank) tests (flexural strength – FS and cycles for failure – CFF), and Weibull modulus with respective 95% confidence intervals for fatigue data (FS and CFF). Results of surface roughness after surface treatments parameters (Ra and Rz in μm – mean and standard deviations) by means of One-way ANOVA and Tukey's post-hoc tests.**

| Groups | Fatigue Data | | Weibull Modulus | | Surface Roughness | |
| --- | --- | --- | --- | --- | --- | --- |
| | Flexural Strength (MPa) | FS | FS | CFF | Ra (μm) | Rz (μm) |
| H-CTRL | 285 (265–305)[B] | 8.04 (5.23–11.47)[A] | 8.04 (5.23–11.47)[A] | 5.84 (3.8–8.35)[A] | 1.80 (0.32)[A] | 11.37 (1.64)[A] |
| L-CTRL | 288 (268–309)[B] | 7.82 (5.14–10.94)[A] | 7.82 (5.14–10.94)[A] | 6.27 (4.11–8.82)[A] | 1.82 (0.29)[A] | 11.66 (1.87)[A] |
| H-AB45 | 310 (292–328)[B] | 9.2 (6.08–12.91)[A] | 9.2 (6.08–12.91)[A] | 7.01 (4.66–9.77)[A] | 1.83 (0.28)[A] | 11.62 (1.90)[A] |
| L-AB45 | 315 (296–333)[B] | 8.86 (5.91–12.24)[A] | 8.86 (5.91–12.24)[A] | 6.78 (4.49–9.42)[A] | 1.87 (0.35)[A] | 12.16 (2.25)[A] |
| H-AB110 | 205 (189–220)[C] | 6.68 (4.46–9.27)[A] | 6.68 (4.46–9.27)[A] | 4.06 (4.69–5.67)[A] | 1.69 (0.35)[AB] | 10.89 (2.20)[A] |
| L-AB110 | 210 (196–225)[C] | 7.58 (5.05–10.52)[A] | 7.58 (5.05–10.52)[A] | 5.07 (3.37–7.04)[A] | 1.78 (0.26)[A] | 11.30 (1.49)[A] |
| H-GLZ | 382 (345–418)[A] | 6.54 (4.19–9.44)[A] | 6.54 (4.19–9.44)[A] | 5.54 (3.56–7.96)[A] | 1.44 (0.31)[BC] | 8.30 (2.14)[B] |
| L-GLZ | 396 (364–430)[A] | 6.80 (4.44–9.63)[A] | 6.80 (4.44–9.63)[A] | 6.09 (3.98–8.62)[A] | 1.23 (0.23)[CD] | 6.83 (1.49)[BC] |
| H-GLZ+HF | 273 (249–297)[B] | 5.98 (3.95–8.37)[A] | 5.98 (3.95–8.37)[A] | 4.86 (3.19–6.87)[A] | 1.38 (0.26)[BC] | 8.15 (1.79)[B] |
| L-GLZ+HF | 280 (256–304)[B] | 6.69 (4.34–9.6)[A] | 6.69 (4.34–9.6)[A] | 5.08 (3.30–7.26)[A] | 0.95 (0.30)[D] | 5.30 (2.35)[C] |

Different capital letters indicate statistical differences for each condition.

In the fatigue data, different capital letters indicate statistical differences for each condition, as determined by the confidence intervals.

| CTRL | AB45 | AB110 | GLZ | GLZ+HF |
|------|------|-------|-----|--------|

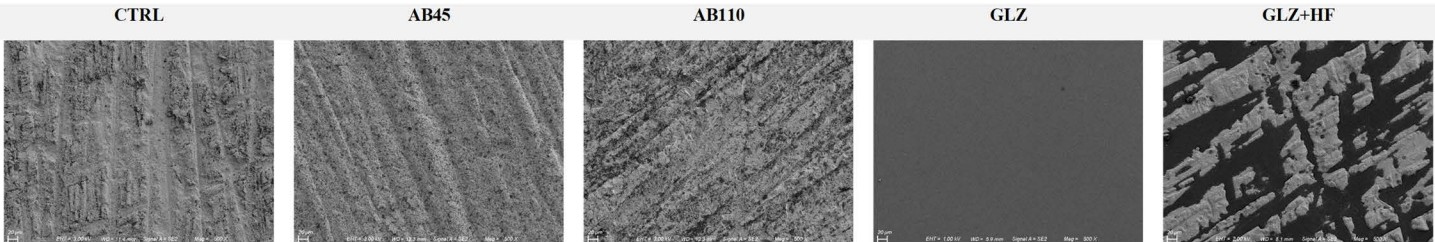

**Fig 4. Representative SEM micrographs of the ceramic after each surface treatment at 500× magnification.** The CTRL group, representing CAD/CAM milling simulation, exhibited the original zirconia surface without any additional treatment. The AB45 group showed smoother abrasion marks. In contrast, the AB110 displayed deeper and more irregular abrasions. The GLZ presented a more uniform and smoother surface, while the GLZ+HF revealed a modified texture with localized roughness.

of 26%. These results demonstrate that air abrasion with smaller alumina oxide particles (45 μm) improves fatigue resistance compared to larger particles (110 μm).

Regarding surface roughness (Ra and Rz parameters), One-way ANOVA shows that surface treatment had a significant effect (Ra: $F = 29.96$; $p < 0.05$ / Rz: $F = 45.86$; $p < 0.05$). GLZ treatments resulted in smoother surfaces with lower Ra and Rz values, with GLZ+HF exhibiting the lowest roughness, followed by GLZ alone (Table 2). In contrast, the AB treatments and the CTRL group showed higher roughness values, with no significant differences among them, indicating that AB was less effective than GLZ in reducing surface defects.

The topographical analysis presented in Fig 5 revealed distinct surface characteristics across the evaluated groups, consistent with their mechanical performance under fatigue. The CTRL group exhibited prominent and continuous milling grooves, serving as the baseline condition without post-processing. AB45 produced a slightly smoother and more uniform surface, effectively reducing superficial defects while preserving surface integrity. In contrast, AB110 indicated the presence of deeper and more irregular subsurface defects, likely acting as stress concentrators and explaining the marked reduction in fatigue resistance. The GLZ group showed smooth and homogeneous surfaces, as the glaze effectively sealed superficial defects, minimizing potential crack initiation sites and enhancing mechanical behavior. When HF etching was applied after GLZ (GLZ+HF), the glaze layer was partially removed, exposing underlying defects and resulting in a heterogeneous surface. Fractographic (Fig 7 and Fig 8) analyses indicate that cracks initiate at the intaglio surface and propagate through the surface of the 4YSZ, influenced by the surface morphology resulting from each surface treatment.

## 4. Discussion

Based on our results, it was observed that the groups treated with GLZ showed a significant improvement in the mechanical performance of the 4YSZ ceramic, while the groups treated with AB110 exhibited a decline in fatigue behavior. Thus, the first hypothesis was accepted. In contrast, the viscosity of the resin cement did not show an impact on the performance of the 4YSZ ceramic, consequently the second hypothesis was rejected. The mechanical response of 4YSZ under fatigue is strongly influenced by the type of surface treatment, confirming that this factor plays a more decisive role than resin cement viscosity.

The fatigue behavior is primarily influenced by prior surface treatment, highlighting that the type of surface treatment plays a critical role in the fatigue behavior of the ceramic according to our findings. As shown in the SEM images (Fig 4), the unmodified surface of the CTRL group and the surface after AB45 treatment present similar points of weakness, resulting in comparable fatigue behavior and interaction with resin cement, with mechanical anchorage ensured by the less prominent defects [25,42]. Although AB is widely used as the preferred surface treatment for zirconia due to its ability to promote mechanical interlocking [43,44], some studies suggest that more aggressive AB parameters may compromise

Table 3. Survival probabilities were assessed across various flexural strength steps and cycle, and their respective standard error measurements.

| Groups | Tension to failure (MPa) / Number of cycles for fatigue failure | | | | | | | | | | | | | | | |
|---|---|---|---|---|---|---|---|---|---|---|---|---|---|---|---|---|
| | 150/ 40X103 | 175/ 50x103 | 200/ 60x103 | 225/ 70x103 | 250/ 80x103 | 275/ 90x103 | 300/ 100x103 | 325/ 110x103 | 350/ 120x103 | 375/ 130x103 | 400/ 140x103 | 425/ 150x103 | 450/ 160x103 | 475/ 170x103 | 500/ 180x103 | 525/ 190x103 |
| H-CTRL | 1 | 1 | 1 | 0.86 (0.08) | 0.73 (0.11) | 0.40 (0.12) | 0.26 (0.11) | 0.13 (0.08) | 0.0 | – | – | – | – | – | – | – |
| L-CTRL | 1 | 1 | 1 | 0.80 (0.10) | … | 0.60 (0.12) | 0.20 (0.10) | 0.06 (0.06) | … | 0.0 | – | – | – | – | – | – |
| H-AB45 | 1 | 1 | 1 | 1 | 0.93 (0.06) | 0.80 (0.10) | 0.33 (0.12) | 0.20 (0.10) | 0.13 (0.08) | 0.0 | – | – | – | – | – | – |
| L-AB45 | 1 | 1 | 1 | 1 | 0.93 (0.06) | 0.80 (0.10) | 0.53 (0.12) | 0.20 (0.10) | 0.06 (0.06) | … | 0.0 | – | – | – | | – |
| H-AB110 | 1 | 0.73 (0.11) | 0.20 (0.10) | … | 0.06 (0.06) | 0.0 | – | – | – | – | – | – | – | – | – | – |
| L-AB110 | 1 | 0.80 (0.10) | 0.40 (0.12) | 0.13 (0.08) | 0.06 (0.06) | 0.0 | – | – | – | – | – | – | – | – | – | – |
| H-GLZ | 1 | 1 | 1 | 0.93 (0.06) | … | … | 0.86 (0.08) | 0.73 (0.11) | 0.60 (0.12) | 0.46 (0.12) | 0.40 (0.12) | 0.26 (0.11) | 0.06 (0.06) | … | 0.0 | – |
| L-GLZ | 1 | 1 | 1 | 1 | 1 | 0.93 (0.06) | … | 0.86 (0.08) | 0.66 (0.12) | 0.46 (0.12) | 0.40 (0.12) | 0.33 (0.12) | 0.13 (0.08) | 0.06 (0.06) | … | 0.0 |
| H-GLZ+HF | 1 | 1 | 0.93 (0.06) | 0.86 (0.08) | 0.40 (0.12) | 0.33 (0.12) | 0.20 (0.10) | 0.13 (0.08) | 0.06 (0.06) | 0.0 | – | – | – | – | – | – |
| L-GLZ+HF | 1 | 1 | 0.93 (0.06) | 0.86 (0.08) | 0.53 (0.12) | 0.40 (0.12) | 0.26 (0.11) | 0.20 (0.10) | 0.0 | – | – | – | – | – | – | – |

The symbol "…" indicates absence of specimen fracturing in the respective step for each condition.

The symbol "–" indicates absence of specimen being tested on the considered step.

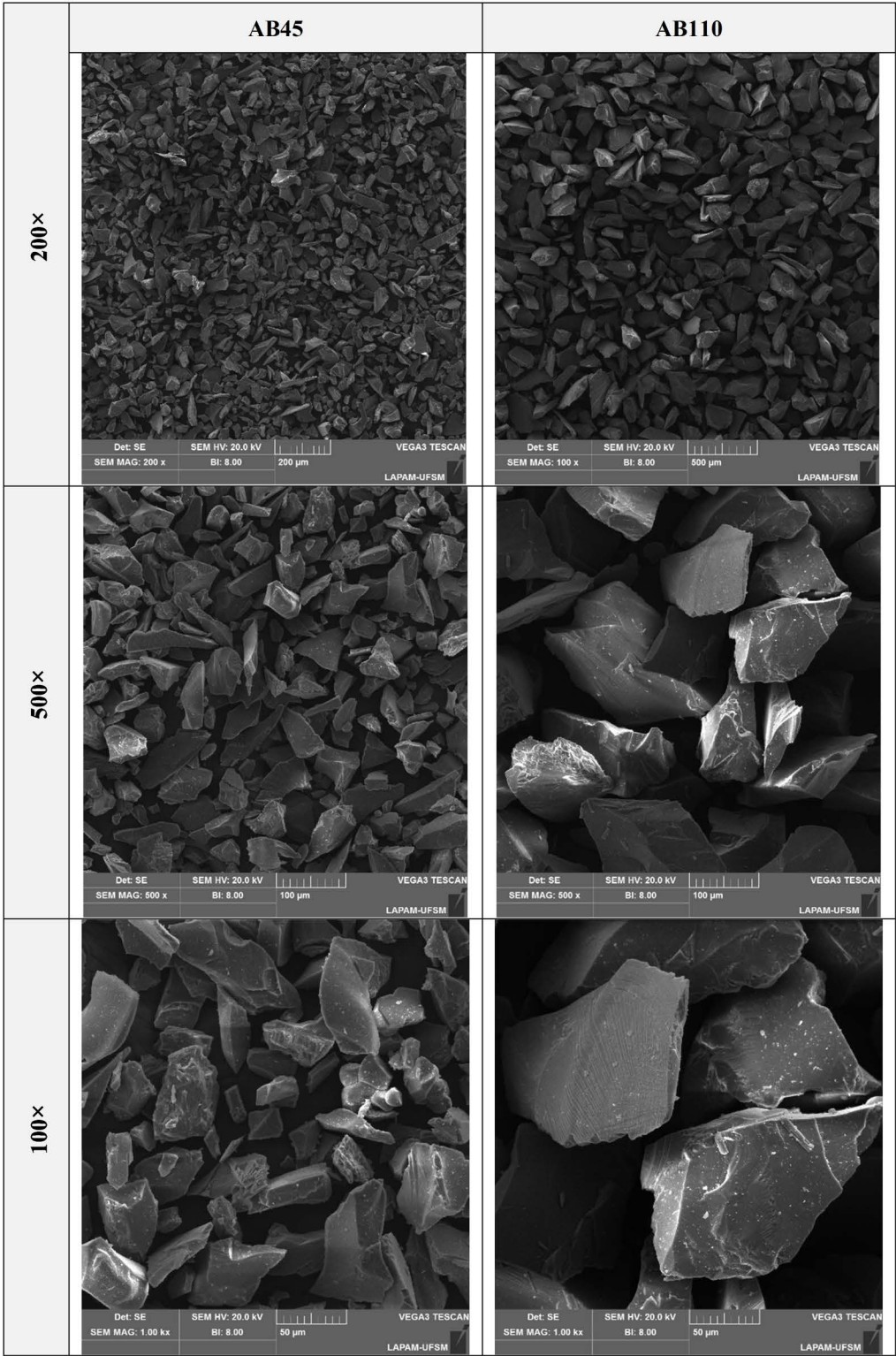

**Fig 5. SEM images of the tested powders with particle sizes of 45 µm and 100 µm at 200×, 500×, and 1000× magnifications.** The 45 µm particles appear smaller and smoother, whereas the 100 µm particles are larger and more aggressive.

the fatigue behavior of 4YSZ [25,45]. The increased cubic phase content and, consequently, the larger grain size in 4YSZ complicate the phase transformation mechanism, making the ceramic more susceptible to failure [5,9,46], necessitating further studies to define optimal parameters and evaluate their consequences on mechanical behavior when new material compositions are developed. Consequently, AB can compromise fatigue behavior under certain conditions [25,45]. Our results corroborate these findings, indicating that AB45 did not significantly compromise mechanical performance compared to CTRL, whereas AB110 induced microcracks originating from pre-existing defects (Fig 4) and was also associated with reduced survival (Table 3).

Building upon the observed differences in fatigue behavior between AB45 and AB110, the influence of abrasive particle size on surface topography and mechanical behavior emerges as a potential contributing factor. An increase in particle size has been associated with the formation of microcracks and a consequent reduction in the material's mechanical properties [47,48]. In general, larger particles during treatment impart greater kinetic energy upon impact [49], resulting in deeper cavities, wider grooves, and more pronounced surface defects [15,19,46,49]. These observations are further supported by Fig 5, SEM images of the tested powders with particle sizes of 45 µm and 100 µm at 200×, 500×, and 1000× magnifications show that the 45 µm particles appear smaller and smoother, whereas the 100 µm particles are larger and more aggressive, corroborating the differences in surface topography and fatigue performance observed between AB45 and AB110. The fatigue data reveals that groups treated with AB45 (L-AB45: 310 MPa; H-AB45: 315 MPa) exhibited significantly higher resistance compared to the AB110 groups (L-AB110: 205 MPa; H-AB110: 210 MPa). The AB110 treatment (L-AB110 and H-AB110) generated notable surface irregularities, characterized by deeper grooves and larger defects relative to AB45, as observed in the SEM images (Fig 4). Such pronounced surface features increase the likelihood of stress concentration points, potentially promoting crack propagation under load and thereby explaining the diminished fatigue performance of the AB110 groups. Conversely, the AB45 treatment produced a more uniform surface, effectively mitigating pre-existing simulation defects and yielding smoother, more consistent morphology, indicating that AB45 does not exert deleterious effects on 4YSZ, with fatigue behavior comparable to the CTRL.

Analysis of the GLZ groups (H-GLZ and L-GLZ) revealed superior fatigue behavior (Table 2). GLZ effectively infiltrates surface defects and porosities, acting as a repair mechanism that produces a more homogeneous surface layer [50]. Recent studies have shown that GLZ application on 4YSZ enhances fatigue behavior by filling pre-existing defects, interrupting crack propagation, and promoting structural stability [25,45]. SEM images (Fig 4) confirm that GLZ results in a smooth, defect-free surface, which potentially reduces stress concentration at critical points and improves mechanical performance (Table 2). Moreover, GLZ treatment significantly reduced surface roughness compared to controls (H-GLZ: 1.44 µm; L-GLZ: 1.23 µm; H-GLZ+HF: 1.38 µm; L-GLZ+HF: 0.95 µm vs. H-CTRL: 1.80 µm; L-CTRL: 1.82 µm), suggesting that it not only covers surface irregularities but also penetrates smaller defects, minimizing initial failure sites [27,45,51].

It is important to note that GLZ and GLZ+HF represent distinct surface modification strategies, resulting in different mechanical outcomes (Table 2). While GLZ alone exhibited superior fatigue behavior (H-GLZ: 382 MPa; L-GLZ: 396 MPa), the GLZ+HF groups performed similarly to controls (H-GLZ+HF: 273 MPa; L-GLZ+HF: 280 MPa), indicating that HF treatment neither improved nor impaired fatigue behavior compared to the control. The acid, designed to selectively etch the vitreous phase of silica-containing ceramics, indiscriminately removed the protective GLZ layer, eliminating the "flaw-healing" effect and re-exposing the inherent defects of the zirconia surface. Consequently, the significant reduction in fatigue behavior observed in this group is a direct result of the protective layer's removal, rendering the material vulnerable to fatigue-induced failures originating from its original surface defects (Fig 4). Overall, while HF selectively modifies the surface, the glaze-only treatment clearly provided superior fatigue behavior, highlighting the benefit of the defect-healing mechanism of GLZ on 4YSZ ceramics (Tables 2 and 3).

Ceramic failure can be influenced by various factors, making it essential to understand the material's behavior at all stages of fabrication [1,20,52]. During the cementation of the prosthetic piece, the cement interacts with the surface treatment and any pre-existing defects on the bonding surface, which can lead to alterations in the ceramic's fatigue behavior

[53]. This process can improve fatigue behavior, as interlocking provides structural reinforcement [53]. The enhancement in the mechanical properties of resin cements is related to their viscosity, which can influence both the seating of the restoration and its handling [54]. More fluid cements can fill defects resulting from previous stages of fabrication and favor a better distribution of stresses. On the other hand, high-viscosity cements contain a higher loading content, which, when there is good integration between the resin matrix and the filler particles, can increase the cement's load-bearing capacity [10,33]. Therefore, the lack of proper adaptation between ceramic surface defects and the resin cement agent can compromise the performance of the restoration [54,55].

Despite the different surface topographies resulting from the treatments, cement viscosity did not influence the behavior of 4YSZ (Table 2 and Fig 6). These results align with recent studies [36], indicating that resin cement viscosity does not

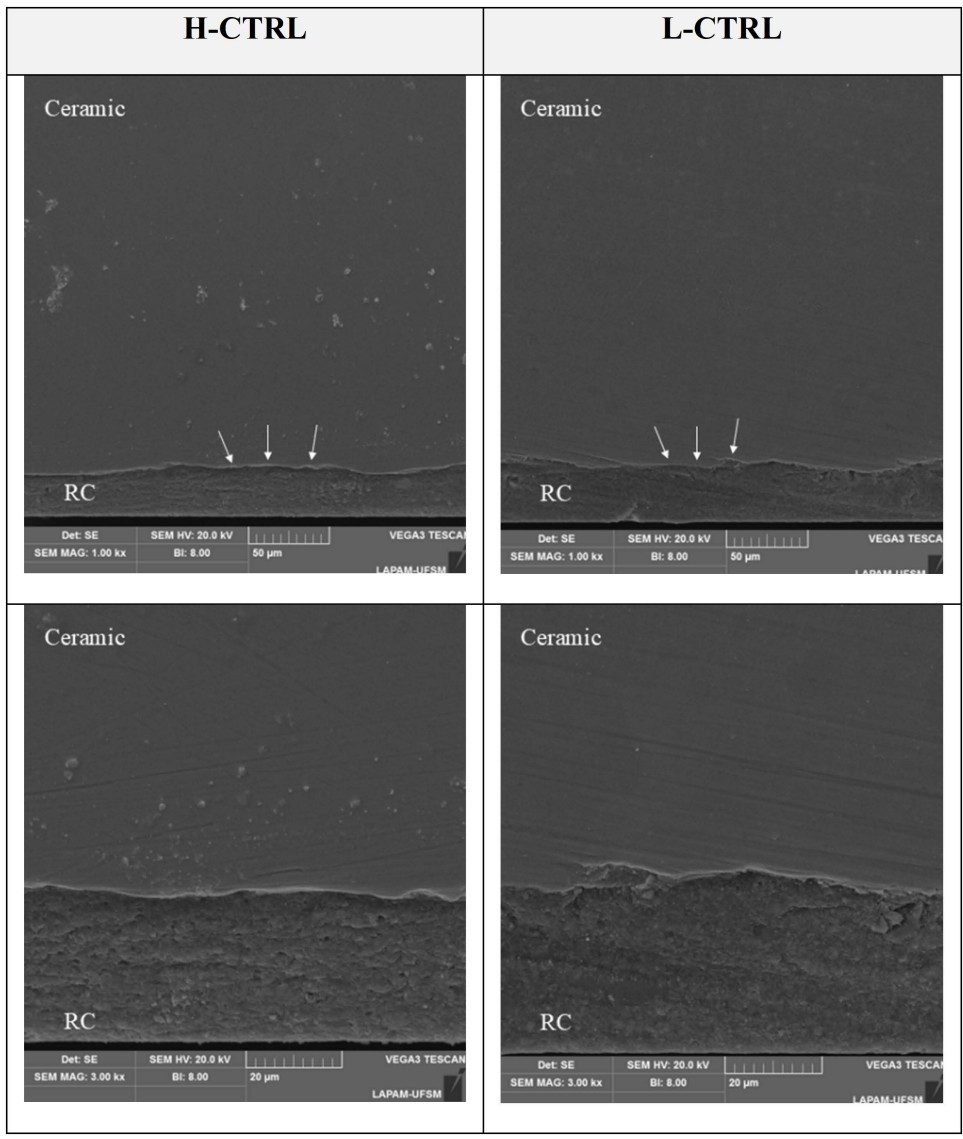

**Fig 6. SEM images of the ceramic-resin cement bonding interface.** The images, at 1000× and 3000× magnification, show the similar penetration of both high-viscosity (H-CTRL) and low-viscosity (L-CTRL) resin cements into the ceramic surface. The elongated depressions, indicated by white arrows, represent the cement's penetration into the microscopic irregularities of the surface, confirming that the cement viscosity did not impede its ability to infiltrate and create a strong bond. The resin cement layer is labeled as RC.

affect the fatigue behavior of 4YSZ ceramic. Although low and high viscosity cements differ, the fatigue behavior of 4YSZ showed no statistically significant differences between the groups (Table 3). Furthermore, Packaeser et al. [39] reported that the elastic modulus and Poisson's ratio, intrinsic properties of resin cements, are similar. The elastic modulus reflects material stiffness and resistance to deformation, while Poisson's ratio describes the relationship between lateral and longitudinal deformations under load, contributing to adaptation to bonding surfaces. With comparable values for these properties, both cement viscosities ensure uniform stress distribution [39], supporting our observation that viscosity does not affect the resistance of 4YSZ ceramic. Consequently, cement viscosity does not alter the effects of surface treatment (Table 2) and results in similar survival rates (Table 3).

Although some groups showed better fatigue behavior than others, the bonding interface analysis revealed that the resin cements regardless of viscosity penetrated ceramic irregularities in a similar manner (Fig 6). These observations support the fatigue data, indicating that resin cement viscosity does not significantly affect the quality of the cementation interface or the fatigue behavior of the 4YSZ ceramic. Therefore, in 4YSZ ceramics, resin cement viscosity appears to have no significant impact, making surface treatments the predominant factor influencing fatigue behavior (Fig 7 and 8).

In summary, the mechanical performance hierarchy observed in our study can be described as follows: GLZ>AB45= CTRL=GLZ+HF>AB110. GLZ treatment provided the most favorable mechanical reinforcement by sealing defects and smoothing the surface. AB45 fatigue behavior comparable to CTRL, providing a smoother and more homogeneous surface than AB110 without introducing damage. AB110 was detrimental, likely due to deeper subsurface defects. HF applied after GLZ partially removed the glaze, leading to intermediate performance similar to CTRL. Resin cement viscosity influenced defect adaptation and stress distribution but did not significantly affect fatigue performance, confirming that surface treatment selection is the primary determinant of 4YSZ fatigue behavior.

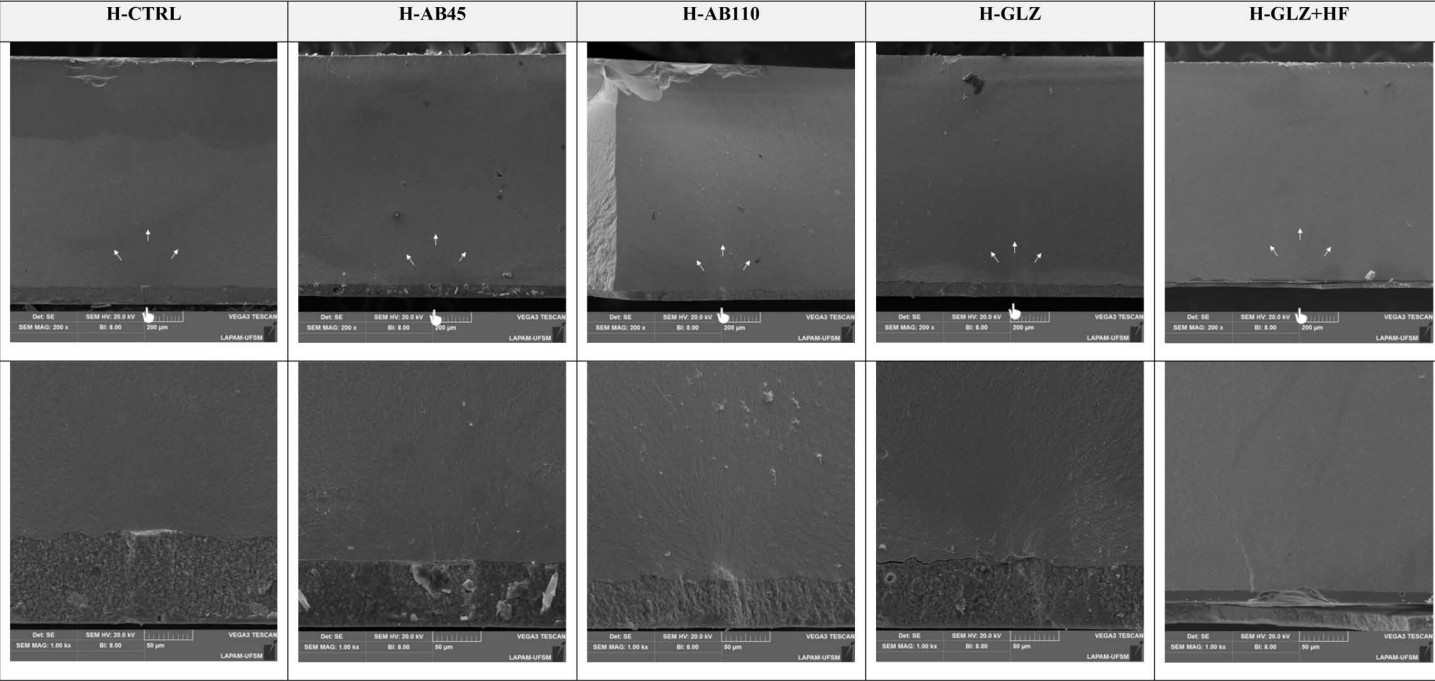

**Fig 7. SEM micrographs of fractured surfaces of high-viscosity cement at 200× and 1000× magnification reveal fractures originating from surface and subsurface flaws (indicated by pointer) located at the center of the ceramic, which corresponds to the tensile side.** The arrows highlight the hackle lines, typical fracture markings.

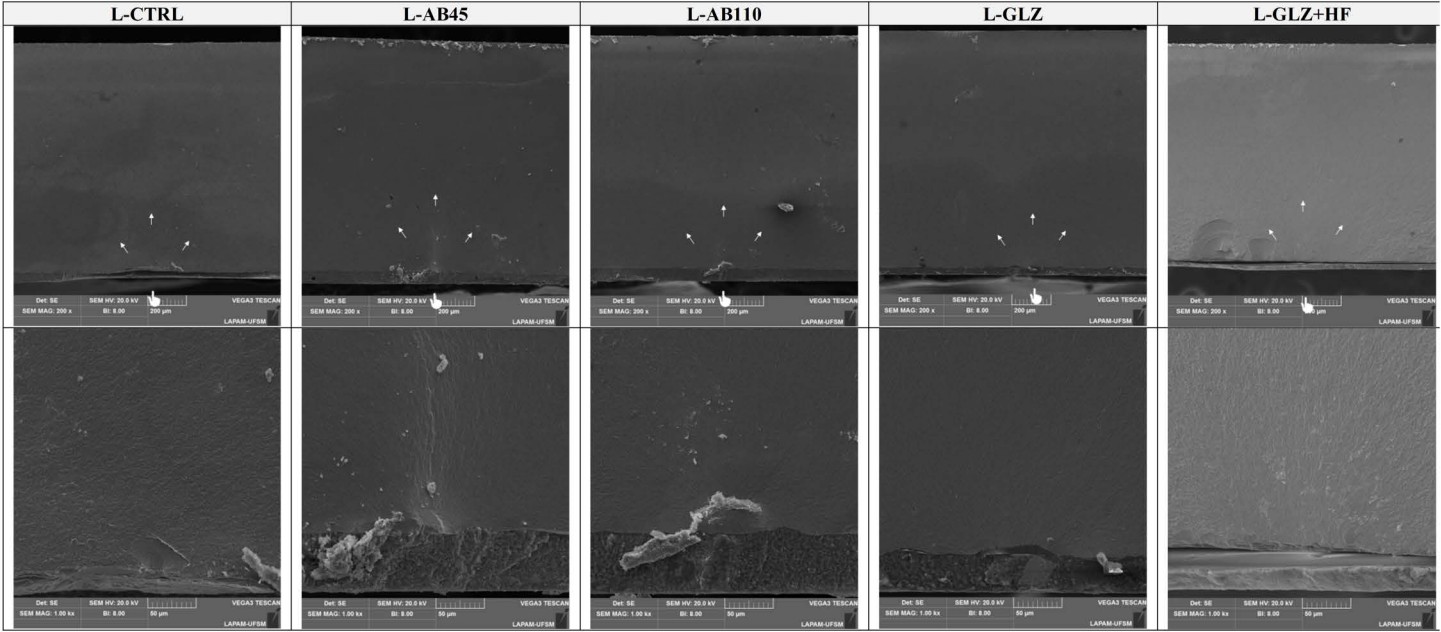

**Fig 8. SEM micrographs of fractured surfaces of low-viscosity cement at 200× and 1000× magnification reveal fractures originating from surface and subsurface flaws (indicated by pointer') located at the center of the ceramic, which corresponds to the tensile side.** The arrows highlight the hackle lines, typical fracture marking.

The study presents important limitations related to both the choice of resin cements and the surface treatments. Only two viscosities of resin cement (high and low) were evaluated, which limits the generalizability of the results, as other cement formulations with different compositions and filler contents may behave differently in terms of adhesion and strength. Similarly, the surface treatments investigated (glaze, with or without hydrofluoric acid etching) represent only a subset of the possible techniques for conditioning 4YSZ ceramics, leaving the effects of other treatments unexplored. Additionally, although Weibull analysis was performed to assess material reliability using a sample size of n = 15 per group, which is considered acceptable, larger sample sizes (e.g., approximately 30 specimens per group) may increase the sensitivity of Weibull parameter estimation. Therefore, the absence of significant differences in Weibull modulus should be interpreted with caution and considered complementary to the survival analysis, which represents the primary outcome of this study. An additional limitation lies in the absence of direct phase transformation analysis, as the mechanistic interpretation is based on previously established literature on zirconia microstructural behavior, including the relationship between surface roughness and mechanical performance. A limitation of this study is that only selected surface treatments with well-established clinical relevance were evaluated, and additional surface modification approaches were not investigated, which may limit the extrapolation of the findings. Furthermore, the application of glaze in a clinical scenario involving anatomically shaped crowns may vary, potentially leading to uneven glaze distribution or areas of over-accumulation, which could cause partial misfit of the restoration and affect its mechanical performance. The results of this study should be interpreted within the limitations of an in vitro design. Although the step-stress fatigue approach is widely used for comparative analysis, it does not fully replicate long-term clinical conditions, and therefore, the findings should not be directly extrapolated to predict clinical longevity

## 5. Conclusions

- The surface treatment methods are more preponderant on the fatigue behavior than variations in resin cement viscosity.

- The use of glaze provided the highest strength among all surface treatments, whereas air abrasion with 110 μm alumina oxide particles resulted in the lowest fatigue behavior, compromising the material's integrity.

- Surface treatments such as air abrasion with 45 μm alumina oxide and glaze followed by hydrofluoric acid etching demonstrated only intermediate outcomes, showing similar fatigue behavior to the control group.

- The viscosities of the resin cements (high and low) did not affect the fatigue behavior of 4YSZ ceramics.

## Supporting information

**S1 Table. Raw dataset of experimental measurements.** Dataset includes groups, RA, RZ, surface treatments, and viscosity values used in the analysis.
(XLSX)

## Author contributions

**Conceptualization:** Kétlin Fagundes Teixeira, Pablo Machado Soares, Gabriel Kalil Rocha Pereira, João Paulo Mendes Tribst, Luiz Felipe Valandro, Ana Carolina Cadore-Rodrigues.

**Data curation:** Kétlin Fagundes Teixeira, Fernanda Cocco, Pablo Machado Soares, Gabriel Kalil Rocha Pereira, Ana Carolina Cadore-Rodrigues.

**Formal analysis:** Kétlin Fagundes Teixeira, Fernanda Cocco, Cornelis Johannes Kleverlaan, João Paulo Mendes Tribst, Luiz Felipe Valandro, Ana Carolina Cadore-Rodrigues.

**Funding acquisition:** Kétlin Fagundes Teixeira, Gabriel Kalil Rocha Pereira, Cornelis Johannes Kleverlaan, Luiz Felipe Valandro, Ana Carolina Cadore-Rodrigues.

**Investigation:** Kétlin Fagundes Teixeira, Fernanda Cocco, Pablo Machado Soares, Gabriel Kalil Rocha Pereira, Cornelis Johannes Kleverlaan, João Paulo Mendes Tribst, Luiz Felipe Valandro, Ana Carolina Cadore-Rodrigues.

**Methodology:** Fernanda Cocco, Pablo Machado Soares, Cornelis Johannes Kleverlaan, João Paulo Mendes Tribst, Luiz Felipe Valandro, Ana Carolina Cadore-Rodrigues.

**Project administration:** Gabriel Kalil Rocha Pereira, Cornelis Johannes Kleverlaan, João Paulo Mendes Tribst, Ana Carolina Cadore-Rodrigues.

**Resources:** Gabriel Kalil Rocha Pereira, Cornelis Johannes Kleverlaan, João Paulo Mendes Tribst.

**Software:** João Paulo Mendes Tribst, Ana Carolina Cadore-Rodrigues.

**Supervision:** Gabriel Kalil Rocha Pereira, Cornelis Johannes Kleverlaan, João Paulo Mendes Tribst, Luiz Felipe Valandro.

**Validation:** Cornelis Johannes Kleverlaan, Ana Carolina Cadore-Rodrigues.

**Visualization:** Kétlin Fagundes Teixeira.

**Writing – original draft:** Kétlin Fagundes Teixeira, Fernanda Cocco, Pablo Machado Soares, Gabriel Kalil Rocha Pereira, Luiz Felipe Valandro, Ana Carolina Cadore-Rodrigues.

**Writing – review & editing:** Kétlin Fagundes Teixeira, Pablo Machado Soares, Gabriel Kalil Rocha Pereira, Cornelis Johannes Kleverlaan, João Paulo Mendes Tribst, Luiz Felipe Valandro, Ana Carolina Cadore-Rodrigues.

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
