## [Decision Letter · Decision Letter 0]

16 Apr 2026

PONE-D-25-54706Biaxial Strength under fatigue of 4YSZ: Surface Treatment and Resin Cement Viscosity effectsPLOS One

Dear Dr. Tribst,

Thank you for submitting your manuscript to PLOS ONE. After careful consideration, we feel that it has merit but does not fully meet PLOS ONE’s publication criteria as it currently stands. Therefore, we invite you to submit a revised version of the manuscript that addresses the points raised during the review process.

Major revision is suggested by the reviewers for the manuscript.

We look forward to receiving your revised manuscript.

Kind regards,

Gaurav Arora, Ph.D

Academic Editor

PLOS One

Journal Requirements:

3. We note that your Data Availability Statement is currently as follows: "

All relevant data are within the manuscript and its Supporting Information files."

4. Please include a copy of Table 5 which you refer to in your text on page 21.

Additional Editor Comments:

Major revision is suggested by the reviewers for the manuscript.

Reviewers' comments:

Reviewer's Responses to Questions

**Comments to the Author**

1. Is the manuscript technically sound, and do the data support the conclusions?

Reviewer #1: Yes

Reviewer #2: Yes

2. Has the statistical analysis been performed appropriately and rigorously? 

Reviewer #1: N/A

Reviewer #2: Yes

3. Have the authors made all data underlying the findings in their manuscript fully available?

Reviewer #1: Yes

Reviewer #2: Yes

4. Is the manuscript presented in an intelligible fashion and written in standard English?

Reviewer #1: Yes

Reviewer #2: Yes

5. Review Comments to the Author

Reviewer #1: 1. The novelty should be more precisely defined. The main contribution is the combined assessment of multiple surface treatments with two resin-cement viscosities on the fatigue biaxial strength of 4YSZ, rather than focusing solely on surface treatment or zirconia bonding in general.

2. The experimental design is interesting, but the use of only two monotonic specimens per condition to define the fatigue parameters is a clear limitation.

3. A stepwise fatigue test involving 10,000 cycles at 20 Hz is effective for initial screening; however, it does not accurately mimic long-term clinical fatigue as constant-amplitude, high-cycle fatigue protocols do.

4. The manuscript could improve by clearly explaining the reasons behind selecting 75 MPa as the initial fatigue stress and the use of 25 MPa increments. While these choices seem to be derived from the monotonic test, the rationale should be articulated more explicitly.

5. The method for determining cement layer thickness has certain limitations. Using a digital caliper to measure thickness indirectly—by subtracting pre- and post-cementation measurements—may not achieve the precision required for a nominal layer of approximately 50 µm, particularly when this variable is a key part of the interpretation.

6. The analysis of the interface that led to the conclusion of 'no viscosity effect' is too limited. The SEM assessment of cement penetration was only conducted for H-CTRL and L-CTRL, making it insufficient to assume that all treated groups exhibit the same interfacial behavior.

Reviewer #2: The manuscript presents an experimental investigation on the fatigue behavior of 4YSZ ceramics subjected to different surface treatments and resin cement viscosities. The topic is relevant to dental materials research, particularly for improving the mechanical reliability of zirconia-based restorations. The study is generally well-structured, and the experimental approach combining fatigue testing, surface characterization, and statistical analysis is appropriate. However, while the work provides useful insights, several methodological, analytical, and presentation-related issues limit its scientific rigor and broader impact.

Although, paper fits well to the submitted journal and, in general, it should be accepted for publication with following major revision suggestions:

1. The authors claim novelty in combining surface treatment with resin cement viscosity effects on 4YSZ. However, the incremental novelty is limited, as: Surface treatment effects on zirconia fatigue are already well established; The study confirms that viscosity has no significant effect, which aligns with existing literature.

2. The manuscript would benefit from clear positioning of its unique contribution, particularly: Why 4YSZ behaves differently from 3Y-TZP in this context; Whether the findings lead to clinically actionable recommendations.

3. The monotonic test sample size (n = 2) is insufficient for reliable estimation of fatigue parameters. This significantly weakens the reliability of the fatigue loading protocol derived from it. The fatigue test uses step-stress accelerated loading, but: No justification is provided for the chosen stress increments and cycle limits; The clinical relevance of 10,000 cycles per step is unclear.

4. The reported 100% statistical power appears unrealistic and likely results from: Overestimated effect size (f ≈ 1.5). Lack of clarity regarding: Whether assumptions for Two-way ANOVA are fully satisfied for fatigue data. Handling of censored data in survival analysis.

5. Weibull analysis is included but not sufficiently interpreted in terms of material reliability.

6. The discussion is largely descriptive and lacks deep mechanistic insight, particularly:

• No quantitative correlation between surface roughness (Ra/Rz) and fatigue strength.

• Limited discussion on crack initiation and propagation mechanisms in 4YSZ.

• The role of phase transformation (tetragonal → monoclinic) is mentioned but not experimentally supported.

7. Several grammatical and structural issues are present: Long sentences in the Introduction and Discussion reduce readability.

8. Some statements are repetitive (e.g., surface treatment dominance). Hypotheses should be more precisely defined.

9. Figures (SEM images) lack: Scale clarity in some cases. Quantitative annotations.

10. The manuscript relies heavily on prior zirconia studies but lacks: Discussion on recent advances in translucent zirconia systems. Comparison with alternative surface treatments (e.g., laser, plasma).

In conclusion, manuscript can be accepted after these suggested major revisions.

6. PLOS authors have the option to publish the peer review history of their article (what does this mean?). If published, this will include your full peer review and any attached files.

Reviewer #1: No

Reviewer #2: **Yes:**Dr. Himanshu Pathak

---

## [Author Response · Author response to Decision Letter 1]

30 Apr 2026

April-27-2026

Manuscript Number: PONE-D-25-54706

Dear Prof. Gaurav Arora

Academic Editor

PLOS One

Thank you for considering our study for publication in the Journal PLOS One. We appreciated the opportunity and revised the manuscript according to the examinations. Comments and responses to the reviewer's suggestions are described below in red and the modifications have been highlighted (in yellow) in the manuscript.

We hope the changes and explanations are adequate. If during this new appreciation more revisions/explanations prove necessary, we are completely at your disposal.

Sincerely,

Dr. Tribst JPM

ACTA -Amsterdam

PONE-D-25-54706

Biaxial Strength under fatigue of 4YSZ: Surface Treatment and Resin Cement Viscosity effects

PLOS One

Dear Dr. Tribst,

Thank you for submitting your manuscript to PLOS ONE. After careful consideration, we feel that it has merit but does not fully meet PLOS ONE’s publication criteria as it currently stands. Therefore, we invite you to submit a revised version of the manuscript that addresses the points raised during the review process.

Major revision is suggested by the reviewers for the manuscript.

• A letter that responds to each point raised by the academic editor and reviewer(s). You should upload this letter as a separate file labeled 'Response to Reviewers'.

As the corresponding author, your ORCID iD is verified in the submission system and will appear in the published article. PLOS supports the use of ORCID, and we encourage all coauthors to register for an ORCID iD and use it as well. Please encourage your coauthors to verify their ORCID iD within the submission system before final acceptance, as unverified ORCID iDs will not appear in the published article. Only the individual author can complete the verification step; PLOS staff cannot verify ORCID iDs on behalf of authors.

We look forward to receiving your revised manuscript.

Kind regards,

Gaurav Arora, Ph.D

Academic Editor

PLOS One

Journal Requirements:

3. We note that your Data Availability Statement is currently as follows: "

All relevant data are within the manuscript and its Supporting Information files."

4. Please include a copy of Table 5 which you refer to in your text on page 21.

We sincerely thank the editor for the consideration of our manuscript. We hope that our work will be considered suitable for publication in PLOS ONE. We would like to confirm that all data required to replicate the results of this study are fully available within the article itself and its Supporting Information files. Specifically, the dataset includes all individual values underlying the reported means, standard deviations, and graphical analyses, ensuring transparency and reproducibility. Therefore, no additional external data sources are required, and the current Data Availability Statement accurately reflects the dataset availability. Regarding the mention of Table 5, this was indeed a typographical error, as no Table 5 is included in the manuscript. This has now been corrected in the revised version. All changes made in the manuscript have been highlighted in yellow for the editor’s and reviewers’ convenience.

Additional Editor Comments:

Major revision is suggested by the reviewers for the manuscript.

Reviewers' comments:

Reviewer's Responses to Questions

Comments to the Author

1. Is the manuscript technically sound, and do the data support the conclusions?

Reviewer #1: Yes

Reviewer #2: Yes

2. Has the statistical analysis been performed appropriately and rigorously?

Reviewer #1: N/A

Reviewer #2: Yes

3. Have the authors made all data underlying the findings in their manuscript fully available?

Reviewer #1: Yes

Reviewer #2: Yes

4. Is the manuscript presented in an intelligible fashion and written in standard English?

Reviewer #1: Yes

Reviewer #2: Yes

5. Review Comments to the Author

Reviewer #1:

1. The novelty should be more precisely defined. The main contribution is the combined assessment of multiple surface treatments with two resin-cement viscosities on the fatigue biaxial strength of 4YSZ, rather than focusing solely on surface treatment or zirconia bonding in general.

2. The experimental design is interesting, but the use of only two monotonic specimens per condition to define the fatigue parameters is a clear limitation.

3. A stepwise fatigue test involving 10,000 cycles at 20 Hz is effective for initial screening; however, it does not accurately mimic long-term clinical fatigue as constant-amplitude, high-cycle fatigue protocols do.

4. The manuscript could improve by clearly explaining the reasons behind selecting 75 MPa as the initial fatigue stress and the use of 25 MPa increments. While these choices seem to be derived from the monotonic test, the rationale should be articulated more explicitly.

5. The method for determining cement layer thickness has certain limitations. Using a digital caliper to measure thickness indirectly—by subtracting pre- and post-cementation measurements—may not achieve the precision required for a nominal layer of approximately 50 µm, particularly when this variable is a key part of the interpretation.

6. The analysis of the interface that led to the conclusion of 'no viscosity effect' is too limited. The SEM assessment of cement penetration was only conducted for H-CTRL and L-CTRL, making it insufficient to assume that all treated groups exhibit the same interfacial behavior.

1. We thank the reviewer for this important comment. The manuscript has been revised to better clarify its novelty. We agree that the main contribution of this study is not the isolated evaluation of surface treatments or resin cement viscosity, but rather the combined effect of multiple surface treatments and different resin cement viscosities on the fatigue behavior of 4YSZ ceramics. In addition, the novelty is reinforced by the fact that 4YSZ presents a distinct microstructural behavior compared to conventional zirconia (3Y-TZP), mainly due to its higher cubic phase content and reduced transformation toughening capacity. Although these differences have been previously described in the literature since the introduction of 4YSZ, several studies have highlighted its distinct mechanical behavior, particularly in relation to surface treatments. In this context, air abrasion, although widely recommended, may introduce surface defects that act as stress concentrators and contribute to crack initiation and propagation, ultimately affecting fatigue performance. Despite this, there is still a lack of studies investigating how different surface treatments interact with resin cement viscosity for this material. This aspect is particularly relevant because resin cement viscosity may influence its ability to penetrate and adapt to surface irregularities created by different treatments, potentially affecting stress distribution and fatigue behavior. Therefore, the interaction between surface topography and cement properties in 4YSZ may differ from previously reported systems. The Introduction section has been revised to better emphasize this combined and material-specific approach. The changes are as follows: "It is essential to investigate how different surface treatments, which produce distinct surface characteristics, interact with resin cement viscosity to determine their combined effect on mechanical performance. The novelty of this study lies in the combined evaluation of multiple surface treatments and two resin cement viscosities on the fatigue biaxial flexural strength of 4YSZ. Unlike previous studies that have predominantly investigated surface treatments or resin cement viscosity separately, this work integrates both factors within the same experimental framework, allowing a direct assessment of their combined effect on mechanical performance. To the best of the authors’ knowledge, no previous study has evaluated the interaction between surface treatment and resin cement viscosity on the fatigue biaxial flexural strength of 4YSZ.”

2. We appreciate the reviewer’s observation. We acknowledge that the use of two specimens for the monotonic test represents a limitation. However, it is important to clarify that the monotonic test in this study was not intended for statistical comparison, but exclusively to provide reference values for defining the fatigue loading protocol. In this context, the monotonic test functions as a screening step, used to estimate the initial load levels to be applied during cyclic fatigue, rather than to generate inferential mechanical data. This approach is well established in the literature on dental materials, where preliminary monotonic results are commonly used to determine a percentage of the failure load for fatigue testing, ensuring that the cyclic loading remains within a clinically relevant and subcritical range. Therefore, the purpose of the monotonic test is methodological, supporting the design of the fatigue experiment, rather than serving as a primary outcome. Additionally, increasing the number of specimens for monotonic testing would not substantially impact the fatigue outcomes, as the fatigue behavior is predominantly influenced by the cyclic loading protocol and material response under repeated stress, which were evaluated with an adequate sample size. This limitation has now been explicitly acknowledged in the manuscript. This approach is consistent with the literature on dental materials, in which preliminary monotonic testing is commonly used to define fatigue parameters based on a percentage of the failure load. Although larger sample sizes may improve statistical reliability, previous studies have adopted relatively small sample sizes for monotonic tests, commonly around n ≈ 5, even when these values are subsequently used to establish fatigue loading conditions (DOI: 10.1016/j.jmbbm.2023.106338; 10.1590/1807-3107bor-2018.vol32.0053; 10.1007/s10266-025-01179-1; 10.1002/jbm.b.35657). This reinforces that monotonic testing in this context is primarily exploratory, rather than inferential. Considering the large number of experimental groups included

---

## [Decision Letter · Decision Letter 1]

5 May 2026

Biaxial Strength under fatigue of 4YSZ: Surface Treatment and Resin Cement Viscosity effects

PONE-D-25-54706R1

Dear Dr. Tribst,

We’re pleased to inform you that your manuscript has been judged scientifically suitable for publication and will be formally accepted for publication once it meets all outstanding technical requirements.

Kind regards,

Gaurav Arora, Ph.D

Academic Editor

PLOS One

Additional Editor Comments (optional):

The reviewers have given acceptance to your manuscript.

Reviewers' comments:

Reviewer's Responses to Questions

**Comments to the Author**

1. If the authors have adequately addressed your comments raised in a previous round of review and you feel that this manuscript is now acceptable for publication, you may indicate that here to bypass the “Comments to the Author” section, enter your conflict of interest statement in the “Confidential to Editor” section, and submit your "Accept" recommendation.

Reviewer #1: All comments have been addressed

Reviewer #2: All comments have been addressed

2. Is the manuscript technically sound, and do the data support the conclusions?

Reviewer #1: Yes

Reviewer #2: Yes

3. Has the statistical analysis been performed appropriately and rigorously? 

Reviewer #1: N/A

Reviewer #2: Yes

4. Have the authors made all data underlying the findings in their manuscript fully available?

Reviewer #1: Yes

Reviewer #2: Yes

5. Is the manuscript presented in an intelligible fashion and written in standard English?

Reviewer #1: Yes

Reviewer #2: Yes

6. Review Comments to the Author

Reviewer #1: The reviewers had addressed all the comments. There are further no comments and the manuscript can be accepted in its current form

Reviewer #2: Authors have revised the paper as per the suggested comments. Now, the paper may be accepted for the publication in this journal.

7. PLOS authors have the option to publish the peer review history of their article (what does this mean?). If published, this will include your full peer review and any attached files.

Reviewer #1: No

Reviewer #2: **Yes:**Dr. Himanshu Pathak

---

## [Editor Report · Acceptance letter]

PONE-D-25-54706R1

PLOS One

Dear Dr. Tribst,

I'm pleased to inform you that your manuscript has been deemed suitable for publication in PLOS One. Congratulations! Your manuscript is now being handed over to our production team.

Kind regards,

on behalf of

Dr. Gaurav Arora

Academic Editor

PLOS One